# Antioxidant and Nitric Oxide Inhibitory Activity of the Six Most Popular Instant Thai Curries

**DOI:** 10.3390/foods13020178

**Published:** 2024-01-05

**Authors:** Sunisa Siripongvutikorn, Kanyamanee Pumethakul, Chutha Takahashi Yupanqui, Vatcharee Seechamnanturakit, Preeyabhorn Detarun, Tanyarath Utaipan, Nualpun Sirinupong, Worrapanit Chansuwan, Thawien Wittaya, Rajnibhas Sukeaw Samakradhamrongthai

**Affiliations:** 1Centre of Excellence in Functional Foods and Gastronomy, Faculty of Agro-Industry, Prince of Songkla University, Hat Yai 90110, Thailand; kanyamanee.p@psu.ac.th (K.P.); chutha.s@psu.ac.th (C.T.Y.); vatcharee.s@psu.ac.th (V.S.); preeya.h@psu.ac.th (P.D.); nualpun.s@psu.ac.th (N.S.); worapanit.c@psu.ac.th (W.C.); 2Department of Science, Faculty of Science and Technology, Prince of Songkla University, Pattani Campus, Rusamilae, Muang, Pattani 94000, Thailand; tanyarath.u@psu.ac.th; 3Center of Excellence in Bio-Based Materials and Packaging Innovation, Faculty of Agro-Industry, Prince of Songkla University, Hat Yai 90110, Thailand; thawean.b@psu.ac.th; 4Division of Product Development Technology, Faculty of Agro-Industry, Chiang Mai University, Chiang Mai 50200, Thailand; rajnibhas.s@cmu.ac.th

**Keywords:** anti-inflammatory, antioxidant, instant food, phytochemical, Thai curry, nitric oxide

## Abstract

All living organisms undergo molecular damage by free radical products. Disrupting the balance between antioxidants and free radicals leads to greater risks of diabetes, hypertension, stroke, and cancer. Consumption of curries containing various herbs and spices provides antioxidant and anti-inflammatory benefits which promote health. The antioxidant and nitric oxide (NO) inhibitory properties of six popular Thai curries, including green curry (G), Panang curry (P), Massaman curry (M), spicy basil leaf curry (SB), southern sour curry (SS), and southern spicy yellow curry (SY) were determined. All six curries contained phenolic and flavonoid compounds and provided antioxidant activity based on electron transfer and hydrogen atom donor properties, as well as having the ability to reduce oxidized metal. The highest antioxidant value was found in SB, followed by M, SS, and SY. The replacement of sugar with dried stevia powder at 50% (Re) improved antioxidant activity. The ORAC assay provided five times higher results than DPPH, ABTS, and FRAP. Extracts of all curries at 1 mg/mL on the macrophage cell line RAW 264.7 showed no cytotoxicity. The highest NO inhibition was found in SB (*p* < 0.05). All curry extracts contained quercetin, kaempferol, luteolin, and apigenin. The six selected popular Thai curries had antioxidant and anti-inflammatory health benefits. Nutraceuticals, functional foods, and the ingredients of each raw material and curry powder should be further investigated.

## 1. Introduction

All living organisms undergo molecular damage by free radical products produced from intrinsic factors such as respiration, cell metabolism, and food catabolism and extrinsic factors including infection, stress, and environmental pollution [1]. Common free radicals produced in the human body are superoxide, hydrogen peroxide, and hypohalite. If the balance of the system fails and not enough antioxidants are supplied, radical products cause chain reactions that lead to cell inflammation and various diseases such as diabetes, hypertension, stroke, and cancer [2]. Antioxidants inhibit free radicals derived from enzyme-producing reactive oxygen species (ROS), including nitric oxide (NO), synthase, and xanthine oxidase, or break down toxic compounds produced by the reaction of free radicals in cells [3]. Antioxidants are generated by endogenous antioxidants such as glutathione peroxide, superoxide dismutase, and catalase [4] and by consumption or supplements in the form of dietary antioxidants including vitamin C, vitamin E, selenium, carotenoid, lycopene, and curcumin [5]. The major dietary antioxidants are vegetables, fruits, herbs, spices, grains, and food supplements. Various dietary antioxidants inhibit free radicals, especially ROS [6]. Many herbs express anti-inflammatory properties through NO inhibition in in vitro and in vivo tests via cytokine and chemokine production. The inhibition of the phosphorylation signaling protein is achieved by stimulating the transcription of the NF-kB gene or inflammatory genes [7].

Thai foods including Massaman curry, Panang curry, green curry, Pad Thai, and Tom Yum are famous worldwide because of their complex tastes, great smell, textural harmony, and flavor. Thai dishes comprise many mixed ingredients in paste or powder form to maximize flavoring, deodorizing, pungency, and coloring to enhance sensory preference. Various herbs and spices contain phytochemical substances with broad and specific medical properties leading to preventive action against some diseases. For example, turmeric-containing curcumin provides balanced blood sugar [8], while alkaloids obtained from black pepper express antitumor, antioxidant, and anti-inflammatory properties [9]. Many curry pastes are classified as raw materials or functional foods [10,11]. To enhance sensory acceptability, salt, sugar, or other seasonings are usually added during cooking. Long-term consumption of high sodium and sugar increases the risk of contracting noncommunicable diseases, including cardiovascular disease, diabetes, hypertension, and kidney disease, especially in the elderly or those with low immunity [12]. Therefore, the reduction or replacement of salt and sugar is prioritized by researchers and consumers worldwide. Potassium chloride (KCl) and stevia help to reduce hypertension and blood sugar, respectively [13,14,15]. However, the effect of antioxidation activities in the replacement of salt and sugar with KCl and stevia powder in curry and seasoning has not been well reported. In addition, the NO inhibitory properties of six popular Thai curries, including green curry, Panang curry, Massaman curry, spicy basil leaf curry, southern sour curry, and southern spicy yellow curry, against the RAW 264.7 cell model using lipopolysaccharide (LPS) as a stimulator [16,17] have not been well studied. Generally, both antioxidant and anti-inflammatory properties have been extensively studied for their association with reducing the incidence of diabetes and hypertension [18,19]. Therefore, this research work aimed to investigate the antioxidant and anti-inflammation activities of high-consumption-ranked Thai curries after reducing NaCl and sugar content by replacement with KCl and stevia powder.

## 2. Materials and Methods

### 2.1. Chemicals and Reagents

The main chemicals and reagents, namely Folin–Ciocalteu’s reagent, were purchased from Loba Chemie PVT.LTD., Mumbai, India. 2,2-diphenyl-1-picryl hydrazyl (DPPH), 2,2-azino-bis-3-ethylbenzthiazoline-6-sulfonic acid (ABTS), 2,4,6-tripyridyl-s-triazine (TPTZ), fluorescein solution, and Lipopolysaccharide (LPS) solution were purchased from Sigma- Aldrich, Darmstadt, Germany. The gallic acid, Trolox, L-Ascorbic acid, Quercetin, and Rutin were HPLC grade and purchased from Sigma-Aldrich, Darmstadt, Germany. 2,2′-azobis(2-methylpropionamidine) dihydrochloride (AAPH) was purchased from FUJIFILM Wako Pure Chemical Corporation, Miyazaki, Japan. Methanol, acetonitrile, and acetic acid were HPLC grade and purchased from RCI Labscan, Bangkok, Thailand, and fetal bovine serum (FBS), penicillin–streptomycin, trypsin-EDTA solution, and RPMI 1640 medium were purchased from Gibco, Waltham, MA USA. 3-(4,5-dimethylthiazol-2-yl)-2,5-diphenyltetrazolium bromide (MTT) was purchased from Invitrogen, Waltham, MA, USA.

### 2.2. Preparation of Fresh Curry Paste

Six popular Thai curries including green curry (G), Panang curry (P), Massaman curry (M), spicy basil leaf curry (SB), southern sour curry (SS), and southern spicy yellow curry (SY) were made using the ingredients shown in Table 1. All ingredients were purchased from a local market in Hat Yai, Songkhla province in Thailand. After grading and washing with chlorine solution 100 ppm at a ratio of 1:3 (ingredient: solution) for 15 min, the gradient was rinsed with tap water 2 times to get rid of excess chlorine residue to lower than 1 ppm. The cleaned ingredients of each curry recipe were blended to a paste before drying using a drum dryer (DD-D12L16, Chareontut, Samutprakarn, Thailand) at 110–120 °C for 2–3 min to obtain dried curry powder with a moisture content of 4–6%. The spicy basil leaf curry was dried in a rotary hot air oven (HS-169, AT Packing, Nonthaburi, Thailand) at 70 °C for 16–18 h to obtain dried curry paste with a moisture content of 4–6%. Each dried sample was ground with a high-speed mixer (WF-20B, Thaigrinder, Thailand) until powder size was lower than 60 mesh (250 µm) (Laboratory test sieve, Endecotts, UK) and coded as a D (dried) sample.

Then, the dried curry powder was added with salt, sugar, and other ingredients based on the recipes in Table 2 and ground using a high-speed mixer to obtain six individual curry powders with particle sizes lower than 60 mesh (250 µm). Each dried curry powder added with salt, sugar, and other ingredients was coded as Or (original).

Salt and sugar contents in each curry powder were replaced by KCl and stevia powder (*Stevia rebaudiana Bertoni*) (STV), respectively (Table 3). Each dried curry powder was blended with seasoning using a high-speed mixer to obtain instant curry powder with a particle size lower than 60 mesh (250 µm). Ratios of salt, sugar, and other ingredients, as well as STV, were based on previous studies and entrepreneur recommendations, approved by 3 expert persons, and reconfirmed by 30 experienced panelists using a 9-point Hedonic scale with approval of the human research ethics committee, Prince of Songkla University (PSU-HREC-2023-008-1-1). Formulas with sensory scores higher than 7.5/9 were selected, adapted, and used in this study. All curries in this step were coded as Re (replacement).

Each curry type was coded as dried curry powder (D), seasoned curry powder (Or), or replacement of salt and sugar with KCl and stevia (Re).

### 2.3. Total Phenolic Content and Antioxidant Activity

#### 2.3.1. Sample Preparation and Extraction

The sample was extracted using the method described by Srisook et al. [20] with some modifications, such as 80% ethanol and 24 h instead of 95% and 5 days. All powders from each curry type and stevia powder (STV) were extracted with 80% ethanol at a ratio of 1:10 (curry powder: 80% ethanol) and stirred in the dark at 25 °C for 24 h. The mixtures were then separated by vacuum suction using a Buchner funnel before centrifuging (CR22GII, Hitachi, Japan) at 4 °C for 20 min at 7100× *g*. Ethanol was completely removed using an evaporator (SB-1000, Tokyo Rikakikai, Tokyo, Japan), and each extract was frozen at −20 °C until use.

#### 2.3.2. Total Phenolic Content (TPC) Determination

TPC was determined using the method described by Singleton et al. [21] with some modifications using a well plate instead of a test tube. Briefly, 20 µL of sample extract was added to a 96-well plate followed by 100 µL of 10% Folin reagent (*v*/*v*). After incubation in the dark at 30 °C for 6 min, 7.5% Na_2_CO_3_ (anhydrous) (*w*/*v*) was added, and the mixture was incubated for another 30 min. The absorbance was measured at 765 nm using a microplate reader (Varioskan LUX, Thermo Scientific, Singapore). TPC was measured using gallic acid, Trolox, and L-ascorbic acid as the standard agents at concentrations of 0–100 µg/mL (R^2^ = 0.999), 0–200 µg/mL (R^2^ = 0.999) and 0–500 µg/mL (R^2^ = 0.999), respectively.

#### 2.3.3. Total Flavonoid Content (TFC) Determination

TFC was determined using the method described by Ha et al. [22]. Briefly, 100 µL of the sample extract was mixed with 100 µL 2% AlCl_3_·6H_2_O (*w*/*v*) and incubated in the dark at 30 °C for 60 min. The absorbance of the mixture was then measured at 420 nm using quercetin and rutin as the standard agents at concentrations of 0–20 µg/mL (R^2^ = 0.999) and 0–80 µg/mL (R^2^ = 0.998), respectively.

#### 2.3.4. DPPH Radical Scavenging Activity

The 2,2-diphenyl-1-picryl hydrazyl (DPPH) radical scavenging activity was determined using the method described by Ding et al. [23]. First, 100 µL of sample extract was mixed with 100 µL 0.2 mM DPPH in 95% ethanol. The mixture was then incubated in the dark for 30 min at 30 °C. Finally, the absorbance was measured at 517 nm using gallic acid, Trolox, and L-ascorbic acid as the standard agents at concentrations of 0–2.5 µg/mL (R^2^ = 0.998), 0–12 µg/mL (R^2^ = 0.998) and 0–14 µg/mL (R^2^ = 0.996), respectively.

#### 2.3.5. ABTS Radical Scavenging Activity

The 2,2-azino-bis-3-ethylbenzthiazoline-6-sulfonic acid (ABTS) assay was determined as described by Arnao et al. [24]. The ABTS radical was generated by incubating 7.4 mM ABTS solution with 2.5 mM K_2_S_2_O_8_ in the dark at 30 °C for 12 h. The radical solution was then diluted to obtain an absorbance of 1.1 ± 0.02 at 734 nm. Then, 20 µL of sample extract was mixed with 280 µL of radical solution and kept in the dark for 2 h at 30 °C. The absorbance of the mixture was measured at 734 nm using gallic acid, Trolox, and L-ascorbic acid as the standard agents at concentrations of 0–22.5 µg/mL (R^2^ = 0.998), 0–110 µg/mL (R^2^ = 0.999) and 0–110 µg/mL (R^2^ = 0.999), respectively.

#### 2.3.6. Ferric Reducing Antioxidant Power (FRAP) Assay

The ferric reducing antioxidant power (FRAP) assay was determined following the method of Benzie et al. [25]. A freshly prepared FRAP solution containing 300 mM acetate buffer pH 3.6, 10 mM 2,4,6-tripyridyl-s-triazine (TPTZ) in 40 mM HCl and 20 mM FeCl_3_·6H_2_O (ratio 10:1:1) was warmed at 37 °C for 30 min. Then, 15 µL of the sample extract was mixed with 285 µL of FRAP solution and incubated for 30 min at 37 °C. The absorbance of the mixture was measured at 593 nm using gallic acid, Trolox, L-ascorbic acid, and FeSO_4_ as the standard agents at concentrations of 0–12 µg/mL (R^2^ = 0.999), 0–100 µg/mL (R^2^ = 0.999), 0–100 µg/mL (R^2^ = 0.999) and 0–90 µg/mL (R^2^ = 0.999), respectively.

#### 2.3.7. Oxygen Radical Absorbance Capacity (ORAC) Determination

The ORAC (oxygen radical absorbance capacity) was determined using the method of Huang et al. [26]. A sample solution of 25 µL was mixed with 150 µL of fluorescein solution 81.6 nM. The mixture was incubated at 37 °C for 15 min and then 25 µL of 2,2′-azobis(2-methylpropionamidine) dihydrochloride (AAPH) 153 mM was added. Fluorescence (excitation wavelength at 485 nm and emission wavelength at 530 nm) was read with 2 min time intervals for 90 min using Trolox as the standard agent at a concentration of 0–170 µg/mL (R^2^ = 0.998).

### 2.4. Phenolic and Flavonoid Profiling Determination by LC-MS

All extract samples were determined for phenolic and flavonoid profiling using LC-MS positive and negative electrospray ionization [27] at the University Center Laboratory with ISO accreditation. Each sample (2 µL) was injected into a Zorbax Eclipse Plus C18 Rapid Resolution HD column (150 mm length × 2.1 mm inner diameter) and performed at 25 °C. The mobile phases were solvent A, a mixture of methanol: acetonitrile: water: acetic acid (10:5:85:1, *v*/*v*), and solvent B, a mixture of methanol: acetonitrile: acetic acid (60:40:1, *v*/*v*) with flow rate 0.2 mL/min. Wavelengths at 230, 280, 368, and 450 nm were used to detect the flavonoid compounds, while 257 and 325 nm were used for phenolic compound determination. Mass spectrometry was run on a Dual AJS ESI for ion source with an MSQ-TOF (model: G6545A, Agilent) and mass spectrometer range of 100–1500 m/z. Electrospray ionization (ESI) was performed when the gas temperature reached 325 °C with a flow of 13 L/min and a Nebulizer at 35 psig for the introduction source. Data were analyzed by MassHunter WorkStation Software Quantitative and Software Qualitative Analysis Workflows V8 with database MassHunter METLIN PCD.

### 2.5. Evaluation of NO Inhibitory Property in Animal Cell Culture

#### 2.5.1. Cytotoxicity of the Macrophage Cell Line RAW 264.7 by MTT Assay

The murine macrophage cell line RAW 264.7 was cultured in RPMI 1640 (Roswell Park Memorial Institute) consisting of 10% fetal bovine serum (FBS) and 100 µg/mL of penicillin–streptomycin with 5% CO_2_ (CO_2_ incubator) at 37 °C. After growth reached 80% confluence of cell tissue culture in the plastic flask, 0.25% trypsin-EDTA solution was used to wash off the culture medium, and the cells were counted at 5×10^5^ cell/mL. Then, 100 µL of cell solution was pipetted into each well of a 96-well plate and incubated for 2 h. RPMI 1640 was replaced with 100 µL of sample solution at a concentration of 1–10 mg/mL and added to each well (using RPMI 1640 100 µL as a standard) and incubated for 24 h.

Thereafter, a 10 µL MTT (3-(4,5-dimethylthiazol-2-yl)-2,5-diphenyltetrazolium bromide) solution at a concentration of 5 mg/mL was added to the 96-well plate and incubated at 37 °C for 2 h. The solution was replaced with 200 µL dimethyl sulfoxide (DMSO) to dissolve the formazan crystals and read at 570 nm. Cell viability was calculated by the equation below, and concentrations higher than 80% were selected to determine anti-inflammatory properties [28].
(1)$$\mathrm{cell}\;\mathrm{viability}\;(\%)=\frac{{O.D.}_{\mathrm{Sample}}}{{O.D.}_{\mathrm{Control}}}\times 100$$
where O.D._Sample_ means absorbance of the sample, and O.D._Control_ means absorbance of the standard (RPMI).

#### 2.5.2. Determination of Extract Anti-Inflammatory Properties by Inhibiting NO Production Using Griess Reagent

The murine macrophage cell line RAW 264.7 was cultured to 80% confluence. After cell growth reached 80%, 0.25% trypsin-EDTA solution was used to wash off the culture medium, and the cells were counted at 5 × 10^5^ cell/mL. Then, 100 µL of cell solution was pipetted into each well of a 96-well plate and incubated for 2 h. Then, RPMI 1640 medium was replaced with 200 µg/mL Lipopolysaccharide (LPS) solution added to each well using RPMI 1640 100 µL as a standard. After that, 100 µL of sample solution at concentrations of 1–10 mg/mL per well was added to the 96-well plate, except for the control and blank, using 100 µL of RPMI and incubated at 37 °C for 24 h. After incubation, 100 µL from each well was transferred to a new 96-well plate for determination of NO inhibition using 100 µL of Griess reagent per well. Spectrophotometry at 570 nm was used to read the 96-well plates. The inhibition of NO production (%) was calculated following the method of Sae-Wong et al. [29]:(2)$$\mathrm{NO}\;\mathrm{inhibition}\;(\%)=\frac{\left({O.D.}_{C}-{O.D.}_{\mathrm{Bc}}\right)-\;({O.D.}_{S}-{O.D.}_{\mathrm{Bs}})}{\left({O.D.}_{C}-{O.D.}_{\mathrm{Bc}}\right)}\times 100$$
where O.D._C_ means absorbance of the control (RPMI + LPS),

O.D._Bc_ means absorbance of the blank (RPMI),O.D._S_ means absorbance of the sample solution (Sample + LPS), andO.D._Bs_ means absorbance of the sample blank (Sample + RPMI).

### 2.6. Statistical Analysis

The experiment was set up using a completely randomized design (CRD). All quality parameters were performed with eight repetitions. Differences in mean values and variations were tested using ANOVA with Tukey’s test (*p* < 0.05). Statistical analysis of the data was carried out using the SPSS statistics software version 22 (IBM, NY, USA).

## 3. Results and Discussion

### 3.1. TPC, TFC, and Antioxidant Activity

TPC, TFC, and antioxidant activity, including the DPPH, ABTS, and FRAP of all samples, was evaluated using gallic acid, Trolox, and ascorbic acid. Only the Trolox equivalent was selected to be present in the graphs because similar trends were found in gallic acid and ascorbic acid.

#### 3.1.1. Total Phenolic Content (TPC)

The TPC values of curry extracts are shown in Figure 1, with the highest ones found in the G, P, M, SB, and SS of D form (*p* < 0.05), followed by curry in the Re and Or samples, respectively. There are no phenolic compounds in bleached salt and sugar. The Or samples were the lowest in TPC (Table 2) because of the dilution effect. Higher TPC contents were found in Re curry samples when 50% sugar was replaced with stevia (Table 3). No significant difference (*p* < 0.05) was found in the TPC of Re-SY and SY due to the low stevia content as a low-sugar recipe (Table 1). Among the various curry types, the highest TPC of SB (Figure 1) was caused by 34% of dried holy basil. This result concurs with Hakkim et al. [30], who reported that the phenolic compound in holy basil leaves (*p* < 0.05) was high and higher than in the inflorescence and stem. Higher contents of carnosic acids, eugenol, and sinapic acid were recorded in holy basil leaves when compared with their inflorescences and stems (*p* < 0.05). Using gallic acid and Trolox as standards, the TPC of SS and SY was in second place after SB. The high TPC of SS and SY was caused by using higher contents of fresh chili and dried chili compared with the other curry recipes. Only SS and SY contained turmeric rhizome, providing antioxidant compounds such as curcumin and its derivatives [31]. Gurnani et al. [32] found that the crude extract of chili (*Capsicum frutescens* L.) seed selected from various local spice markets in India contained a wide range of phenolic and flavonoid compounds as 7.95–26.15 GAE mg/g DW extract and 4.64–12.84 RU mg/g DW extract, respectively, and reported that there were also many biologically important volatile constituents, including heterocyclic compounds, β-diketones, hydrocarbons, long-chain aliphatic carboxylic acids, etc.

The M, G, and P curry mixtures (Table 1) showed that adding more mixed spices gave a higher TPC (Figure 1); therefore, M exhibited the highest TPC. Lu et al. [33] reported that chlorogenic acid and rutin were major compounds found in the extracts from 18 common spices used in Chinese cuisine, including star anise, fennel, cumin, angelica dahurica root, green prickleyash, Sichuan pepper, dried tangerine peel, white pepper, nutmeg, galangal, dried ginger, tsaoko amomum fruit, villous amomum fruit, dried chili pepper, bay leaf, cinnamon, mustard, and curry powder. Phenolic compounds also play a key role in antioxidant activity [33].

The three standards used in this experiment were gallic acid (Appendix A), Trolox (Figure 1), and ascorbic acid (Appendix A). Because of the similar trend in the standards used, only Trolox was selected to be explained. The highest TPC was found in the ascorbic acid equivalent, followed by the Trolox equivalent and gallic acid equivalent. Ascorbic acid is also known as vitamin C, a good external antioxidant found in the body [34]. Therefore, consumption of the six selected curries or their food products improved the antioxidant activity of various phytochemicals comparable to high amounts of vitamins C and E or Trolox (vitamin E equivalent).

#### 3.1.2. Total Flavonoid Content (TFC)

Quantity and changes in TFC in the six selected curries corresponded to TPC (Figure 2). Generally, dried curries (D) contained TFC significantly higher than seasoned curries (Or) (*p* < 0.05). There was lower TFC content in the added salt and sugar sample (Or) due to the dilution effect, as described earlier. SS and SY contained higher TFC than G, P, M, and SB, caused by the turmeric rhizome, which contains curcumin, a potent flavonoid with various phytochemical properties [35]. Curries with sugar replacement by stevia (Re) had significantly higher TFC than the Or samples (*p* < 0.05). Extracts with stevia contained TFC at 1969.93 ± 158.93 µg QE/g DW and 6971.81 ± 564.36 µg RE/g DW, which is higher than all other curry extracts (*p* < 0.05) (Figure 2). Various TFC components were found in *Stevia nepetifolia*, including apigenin-40-O-glucoside, luteolin-7-O-glucoside, kaempferol-3-O-rhamnoside, quercitrin, quercitin-3-O-glucoside, and quercetin-3-O-arabinoside [36]. The replacement of sugar with stevia significantly improved TFC (*p* < 0.05), especially in Re-M and Re-SY. TFC increased at higher mixed spice contents, with the TFC of M and G being higher than P, as observed by Lu et al. [33].

Similar TFC trends were also found in the two flavonoid standard agents (quercetin (Appendix A) and rutin (Figure 2)). Generally, using rutin as a standard equivalent agent provided higher TFC than the quercetin equivalent. Rutin exhibited a lower reaction ability to TFC than quercetin. Quercetin and rutin are major plant flavonoids, providing effective antioxidant and anti-inflammatory properties [37].

### 3.2. Phytochemicals Profiling by LC-MS

The results of phenolic and flavonoid profiling by LC-MS revealed the presence of 17–54 flavonoid compounds, 10–23 phenolic compounds, 8–14 terpenes, and 14–21 other compounds (e.g., quinone, alkaloid, chromones, etc.) in six curry extracts (data in publication process). Major phenolic acids identified in these extracts were quinic acid, shikimic acid, gallic acid, glucocaffeic acid, and transcinnamic acid. Important flavonoids detected were quercetin, kaempferol, luteolin, and apigenin (Figure 3 and Appendix A). In addition, the SS and SY extracts showed the presence of curcumin, a good flavonoid derived from turmeric, with highly beneficial effects such as anti-inflammation, anticancer, DNA protective dietary compounds, and cardiovascular benefits [38,39]. When considering the raw materials used for making the curry extracts, the extracted M contained the greatest number of spices and herbs, totaling 16 items, which was more than other types of curry. The higher varieties contributed to M having the highest number of flavonoid compounds. However, the total flavonoid content (TFC) of M was lower when compared with SS and SY. This pointed out that quality (number of compounds) may not relate well to quantity (content), and even though various dried items [33] were used, this may be due to compound deterioration during the drying process. In addition, SB exhibited significantly higher values of total phenolic content (TPC) and antioxidant activities compared with other samples (*p* < 0.05). Based on the LC-MS results of phenolic and flavonoid profiling, Cynaroside A (Figure 3) from spicy holy basil leaf curry was identified as a key compound suspected to contribute to strong antioxidant effects by inhibiting electron transfer and radical adduct formation activities [40,41,42]. It was hypothesized that the Cynaroside A compound may have more beneficial health effects that should be further investigated and utilized, particularly if there is an increase in content.

### 3.3. Antioxidant Activity (DPPH, ABTS, FRAP, and ORAC Assays)

The antioxidant activities of the six selected curry extracts were determined using the DPPH, ABTS, FRAP, and ORAC assays, with gallic acid equivalent (Appendix A), Trolox (Figure 4, Figure 5, Figure 6 and Figure 7) equivalent, and ascorbic acid equivalent (Appendix A). The FRAP assay used FeSO_4_ as a standard agent (Appendix A). The electron and hydrogen atom transfer of the antioxidant to free radical [43] were determined by the DPPH (Figure 3) and ABTS methods (Figure 5). Each dried curry extract (D) exhibited the highest antioxidant activities, followed by the Re and Or samples. Higher stevia replacement induced higher antioxidant activity, particularly in Re-SS and Re-SY (Table 2 and Table 3). The ABTS assay indicated a similar mechanism to DPPH for electron transfer and hydrogen atom donor properties, but ABTS has broad spectrum molecular polarity suiting wider environmental conditions compared with DPPH [44]. Therefore, antioxidant activity determined by ABTS was higher than the DPPH assay [45]. Assay choice was one of the determinant factors of antioxidant activity not only for antioxidant compounds but also for solvent extracts involving polarity and standard agents [43]. Electron transfer capacity to the ferric ion of the antioxidant was determined by the FRAP assay (Fe^3+^ was changed to Fe^2+^) (Figure 6). The FRAP assay expressed activities of the six curry powders, with similar trends to the DPPH and ABTS assays. Interestingly, the antioxidant capacity of the SB extract determined by FRAP was the highest, followed by SS > M > SY > G > P (*p* < 0.05) because dried holy basil was only used in SB. Chaudhary et al. [46] reported that holy basil leaves (*O. sanctum*) showed a strong transfer hydrogen atom ability to free radicals and inhibited other subsequent damages due to 16 phenolic compounds detected by LC-MS assay, including apigenin, vitexin, vicenin 2, rosmarinic acid, caffeic acid, chlorogenic acid, etc. The DPPH and ABTS of the SS and M extracts provided antioxidants in second place after SB. The highest antioxidant capacity of SB based on all in vitro tests, including DPPH, ABTS, and FRAP, was due to holy basil leaves. The other curry powders showed diverse antioxidant activity for different assays. For example, G, P, and SY were significantly different (*p* < 0.05) based on the DPPH assay, while no significant difference was found using the ABTS assay due to antioxidant polarity deriving from the extract, solvent used, and radical preparation. The DPPH assay used the 2,2-diphenyl-1-picryl hydrazyl radical to determine lower polarity molecules compared with the ABTS assay which used the 2,2-azino-bis-3-ethylbenzthiazoline-6-sulfonic acid radical [47,48]. Based on three standards, namely gallic acid, Trolox, and ascorbic acid, the antioxidant activities of each extract followed the same trend. The ascorbic acid equivalent provided the highest value using the DPPH and FRAP assays, while the Trolox equivalent was the highest standard by the ABTS assay, confirming that molecular polarity was the determinant factor for antioxidant activity with higher compatibility using reduced content.

The ORAC assay is used to determine antioxidant properties based on an in vivo system of the hydrogen atom transfer (HAT) reaction toward peroxyl radicals as the most abundant free radicals in human cells [49]. Antioxidant activity was found in all six selected curries (Figure 7), especially in SB, which had the highest antioxidant activity (*p* < 0.05), followed by the M ≈ P ≈ G ≈ SY samples (nonsignificant, *p* < 0.05) and SS, respectively. Lu et al. [50] stated that six spices, including garlic, onion, red chili, paprika, ginger, and black pepper powder, in meat products exhibited a high ORAC assay based on both electron transfer and hydrogen donation. All spices provided inhibitory effects for developing heterocyclic amines (HCAs), with the formation of polycyclic aromatic hydrocarbons (PAHs) mainly inhibited by electron transfer ability. The reduction in carcinogen agents derived from HCAs and PAHs, commonly found in deep-fried meatballs, was recorded when some spices were applied [50].

Using stevia provided higher antioxidant activity in the Re samples determined by the ORAC assay compared with no stevia content (Or). ORAC results obtained from Re samples show similar trends to the other assays. The ORAC assay of stevia extract was the highest among all other extracts (*p* < 0.05). The ORAC assay exhibited a strong relationship with TFC (R^2^ > 0.85) because stevia contains phytochemicals, phenolic acids, flavonoids, and antioxidant activity. This result agreed with the finding of Lemus-Mondaca et al. [51], who reported increased antioxidant activity in dried stevia using freezedrying, convective drying, vacuum drying, microwave drying, sun drying, and shade drying with 17% (FRAP) and 40% (ORAC) compared with fresh leaves [50]. Results indicate that phytochemicals providing the biological properties and antioxidant activities of stevia were heat stable and/or generated during the drying process. Therefore, the drying process of some raw materials not only provides shelf-life stability but also reduces product weight and enhances biological activity.

### 3.4. Cytotoxicity and NO Inhibitory Activity

#### 3.4.1. Cytotoxicity Test Using the Macrophage Cell Line RAW 264.7 Model

The cytotoxicity of the six selected curries, including G, P, M, SB, SS, and SY, and stevia extract at concentrations of 10, 5, and 1 mg/mL is shown in Figure 8. More than 80% confluence of the cells was obtained after treatment with all the curry extracts at concentrations of 10, 5, and 1 mg/mL, except for the stevia extract. A higher interference from the color of all extracts was found when a higher concentration was used (10 and 5 mg/mL). Therefore, the concentration at 1 mg/mL was selected to determine the inhibitory effect on NO production. G, P, M, SB, and SY at concentrations of 1 mg/mL enhanced RAW 264 macrophage cell growth with M, SS, and SY, respectively. The toxicity test was conducted to determine the NO inhibition effect. Higher cell proliferation for cell recovery should also be further studied.

#### 3.4.2. Inhibition of NO Production

Free radicals in the human body as ROS are generated by cell metabolism. Higher ROS produces higher NO [52,53,54], leading to inflammation and cell molecular damage. Therefore, the reduction or inhibition of NO production is an essential parameter for inflammation and antioxidant determination. The highest NO production inhibition of SB was found at 68.86 ± 0.18, followed by STV, P, SY, G, SS, and M, respectively (Figure 9). The inhibition of NO production in each selected curry sample depended on its antioxidant activity. Among the tested curry samples, SB had the highest values of TPC, TFC, and all tested antioxidant assays. The results of the LC-MS chromatogram indicated four major flavonoids in the six curry extracts, including quercetin, kaempferol, luteolin, and apigenin (shown in Appendix A; Figure 3 and Appendix A), concurring with Tuntipopipat et al. [55], who reported that the activities of the four flavonoids and three carotenoids against NO production in the RAW cell line. Interestingly, there were many compounds found in the spicy basil leaf curry without identification that should be further studied (Figure 3).

## 4. Conclusions

The six popular Thai curries used in this study were green curry (G), Panang curry (P), Massaman curry (M), spicy basil leaf curry (SB), southern sour curry (SS), and southern spicy yellow curry (SY). Replacement with KCl and stevia leaf powder in all curries with a proper ratio of 50% for reducing NaCl and sugar content provided a sensory acceptability over 7.5/9. All curry powders contained phytochemicals, including phenolics, flavonoids, terpene, and others in the range 10–23, 17–54, 8–14, and 14–2, respectively. Only the extracted SB contained Cynaroside A, indicating very strong antioxidant and NO inhibitory properties among other curry powders. SB had the highest total phenolic contents and antioxidant activities, followed by M, SS, and SY. Using the ascorbic acid equivalent gave the highest values, followed by the Trolox and gallic acid equivalents, respectively. Dried stevia leaf powder was the greatest ingredient yielding the highest total flavonoid content. The replacement of sugar with dried stevia at 50% increased antioxidants in the Re samples. Using the ORAC assay for antioxidant determination gave results five times higher than the DPPH, ABTS, and FRAP assays. Cytotoxicity testing on the macrophage cell line RAW 264.7 showed that at a concentration of 1 mg/mL, all tested extract samples had a cell viability of more than 80%. The SB sample showed the highest NO inhibition (*p* < 0.05). Based on the LC-MS analysis, the six selected curry extracts contained at least four flavonoids, including quercetin, kaempferol, luteolin, and apigenin, giving high potential activity for biological effects. Therefore, the six selected popular Thai curries are good healthy foods due to their antioxidant and NO inhibitory effects. This scientific data should be made available to consumers, industries, farmers, and those involved in the supply chain. Nutraceuticals, functional food, and the ingredients of each raw material and curry powder should be further investigated to open other utilization platforms.

## Figures and Tables

**Figure 1 foods-13-00178-f001:**
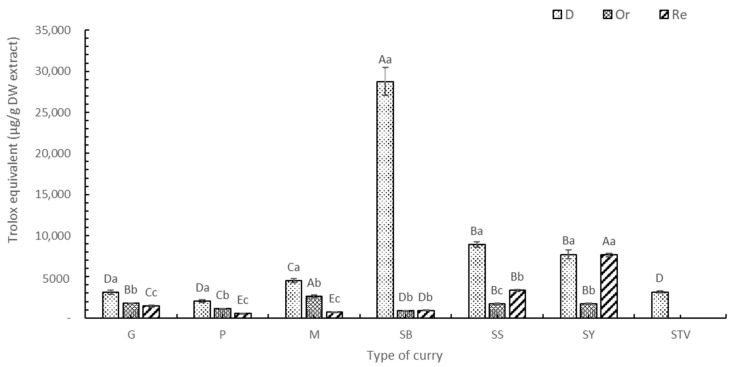
TPC of curry and stevia extracts using Trolox as a standard. G means green curry, P means Panang curry, M means Massaman curry, SB means spicy basil leaf curry, SS means southern sour curry, SY means southern spicy yellow curry, STV means stevia powder, D means dried curry (pure), Or means seasoned curry, and Re means curry with sugar replaced by stevia at 50%. Different uppercase letters indicate significant differences between each curry treatment, i.e., G, P, M, SB, SS, SY, and STV. Different lowercase letters indicate significant differences between treatments within the same type of curry, such as D, Or, and Re (*p* < 0.05).

**Figure 2 foods-13-00178-f002:**
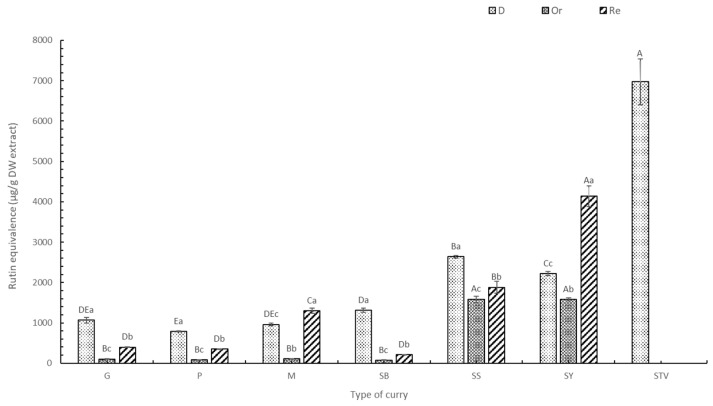
TFC of curry and stevia extracts using rutin as a standard. G means green curry, P means Panang curry, M means Massaman curry, SB means spicy basil leaf curry, SS means southern sour curry, SY means southern spicy yellow curry, STV means stevia powder, D means dried curry (pure), Or means seasoned curry, and Re means curry replacing sugar with stevia at 50%. Different uppercase letters indicate significant differences between each curry treatment, i.e., G, P, M, SB, SS, SY, and STV. Different lowercase letters indicate significant differences between treatments within the same type of curry, such as D, Or, and Re (*p* < 0.05).

**Figure 3 foods-13-00178-f003:**
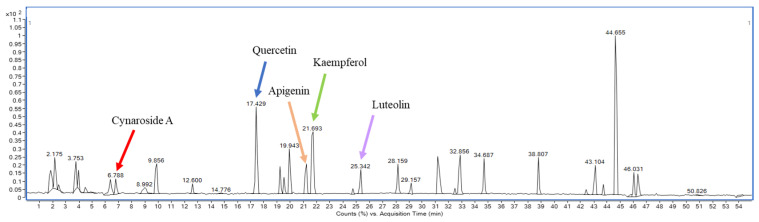
Chromatogram of phenolic compounds found in spicy basil leaf curry (SB) determined by LC-ESI-QTOF-MS/MS.

**Figure 4 foods-13-00178-f004:**
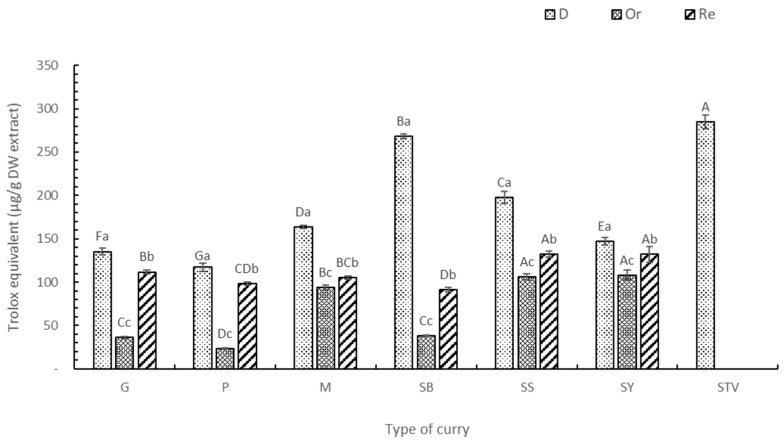
Antioxidant activity by the DPPH assay of curry and stevia extracts using Trolox as a standard. G means green curry, P means Panang curry, M means Massaman curry, SB means spicy basil leaf curry, SS means southern sour curry; SY means southern spicy yellow curry, STV means stevia powder, D means dried curry (pure), Or means seasoned curry, and Re means curry replacing sugar with stevia at 50%. Different uppercase letters indicate significant differences between each curry treatment, i.e., G, P, M, SB, SS, SY, and STY. Different lowercase letters indicate significant differences between treatments within the same type of curry, i.e., D, Or, and Re (*p* < 0.05).

**Figure 5 foods-13-00178-f005:**
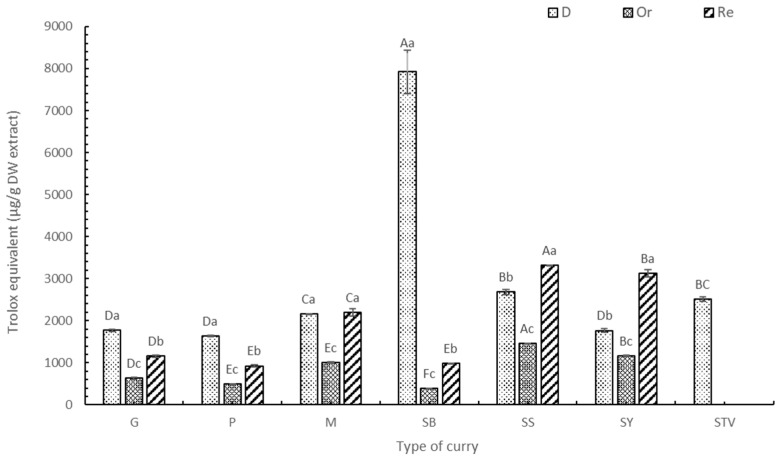
Antioxidant activity by the ABTS assay of curry and stevia extracts using Trolox as a standard. G means green curry, P means Panang curry, M means Massaman curry, SB means spicy basil leaf curry, SS means southern sour curry, SY means southern spicy yellow curry, STV means stevia powder, D means dried curry (pure), Or means seasoned curry, and Re means curry replacing sugar with stevia at 50%. Different uppercase letters indicate significant differences between each curry treatment, i.e., G, P, M, SB, SS, SY, and STY. Different lowercase letters indicate significant differences between treatments within the same type of curry, i.e., D, Or, and Re (*p* < 0.05).

**Figure 6 foods-13-00178-f006:**
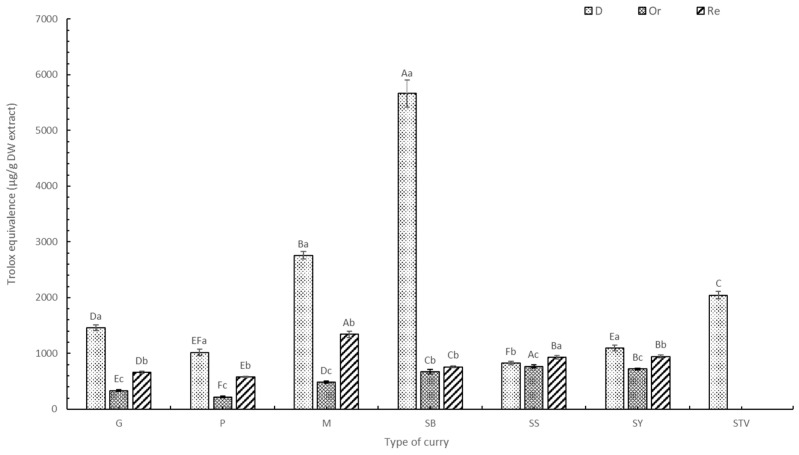
Antioxidant activity by the FRAP assay of curry and stevia extracts using Trolox as a standard. G means green curry, P means Panang curry, M means Massaman curry, SB means spicy basil leaf curry, SS means southern sour curry, SY means southern spicy yellow curry, STV means stevia powder, D means dried curry (pure), Or means seasoned curry, and Re means curry replacing sugar with stevia at 50%. Different uppercase letters indicate significant differences between each curry treatment, i.e., G, P, M, SB, SS, SY, and STY. Different lowercase letters indicate significant differences between treatments within the same type of curry, i.e., D, Or, and Re (*p* < 0.05).

**Figure 7 foods-13-00178-f007:**
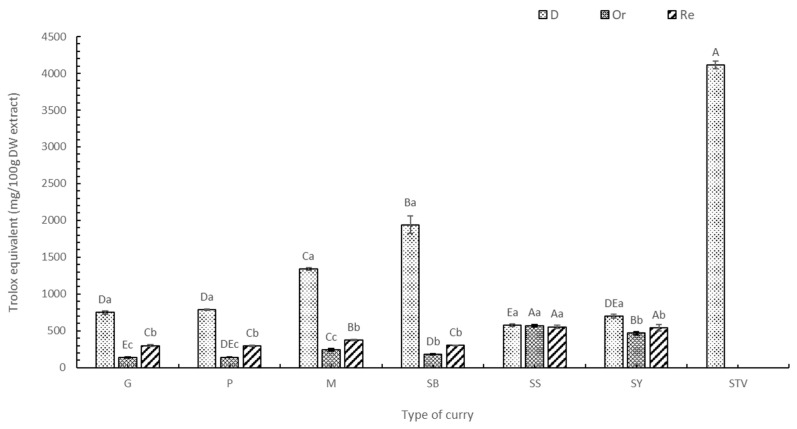
Antioxidant activity by the ORAC assay of curry and stevia extracts using Trolox as a standard. G means green curry, P means Panang curry, M means Massaman curry, SB means spicy basil leaf curry, SS means southern sour curry, SY means southern spicy yellow curry, STV means stevia powder, D means dried curry (pure), Or means seasoned curry, and Re means curry replacing sugar with stevia at 50%. Different uppercase letters indicate significant differences between each curry treatment, i.e., G, P, M, SB, SS, SY, and STY. Different lowercase letters indicate significant differences between treatments within the same type of curry, i.e., D, Or, and Re (*p* < 0.05).

**Figure 8 foods-13-00178-f008:**
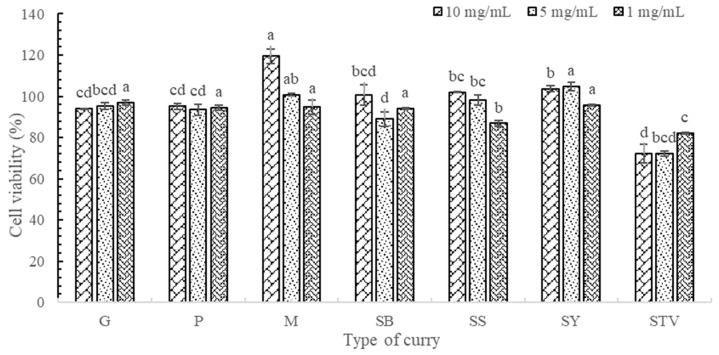
Cell viability of curry and stevia extracts. Different lowercase letters indicate significant differences within the same treatment groups (*p* < 0.05). G means green curry, P means Panang curry, M means Massaman curry, SB means spicy basil leaf curry, SS means southern sour curry, SY means southern spicy yellow curry, and STV means stevia powder.

**Figure 9 foods-13-00178-f009:**
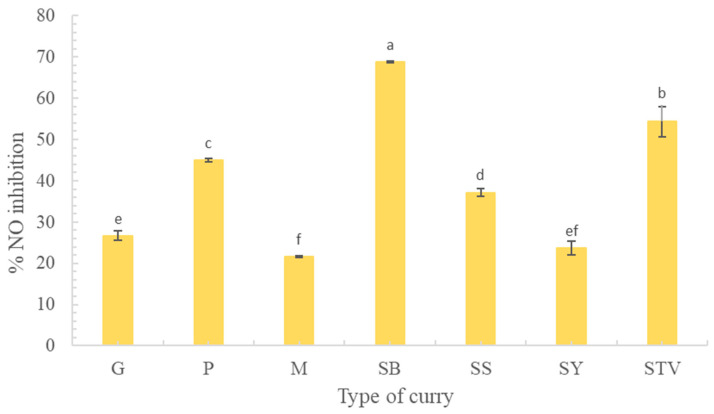
Nitric oxide (NO) inhibition of each curry type and stevia extract. Different lowercase letters indicate significant differences within the same treatment group (*p* < 0.05). G means green curry, P means Panang curry, M means Massaman curry, SB means spicy basil leaf curry, SS means southern sour curry, and SY means southern spicy yellow curry.

**Table 1 foods-13-00178-t001:** Ingredients used to make the six Thai curries.

Ingredients	Amount (%)					
	G	P	M	SB	SS	SY
Fresh lemongrass (*Cymbopogon citrates* L.)	30	10	5	-	-	40
Fresh green chili (*Capsicum annuum* L.)	35	-	-	5.5	-	-
Fresh galangal (*Alpinia galanga* L.)	5	10	5	-	-	-
Fresh shallot bulb (*Allium ascalonicum* L.)	15	30	35	-	5	5
Fresh garlic bulb (*Allium sativum* L.)	10	25	15	45	10	10
Dried chili pepper (*Capsicum annuum* L.)	-	20	15	-	-	-
Dried black pepper (*Piper nigrum* L.)	-	-	1	7	-	5
Fresh ginger (*Zingiber officinale Roscoe*.)	-	-	4	-	-	-
Fresh coriander root (*Coriandrum sativum* L.)	-	-	-	3	-	-
Fresh red chili (*Dendranthema indicum* L.)	-	-	-	5.5	30	7
Dried chili (*Dendranthema indicum* L.)	-	-	-	-	40	25
Fresh turmeric (*Curcuma longa* L.)	-	-	-	-	15	5
Mixed spices	5	5	20	-	-	3
Dried holy basil leaves (*Ocimum sanctum* L.)	-	-	-	34	-	-

G means green curry, P means Panang curry, M means Massaman curry, SB means spicy basil leaf curry, SS means southern sour curry, and SY means southern spicy yellow curry.

**Table 2 foods-13-00178-t002:** Ingredients used for making each original instant curry powder (Or).

Ingredient	Amount (%)					
	G	P	M	SB	SS	SY
Dried curry powder	33	45	33	13	80	64
Mixed spices	4	-	-	-	-	5
Salt	13	15	12	24	17	22
Sugar	50	50	55	59	3	9
Dried basil leaves	-	-	-	4	-	-

G means green curry, P means Panang curry, M means Massaman curry, SB means spicy basil leaf curry, SS means southern sour curry, and SY means southern spicy yellow curry.

**Table 3 foods-13-00178-t003:** Instant curry powder ingredients with the replacement of salt and sugar with KCl and stevia (Re).

Ingredient	Amount (%)					
	G	P	M	SB	SS	SY
Dried curry powder	46	45	43	19	80	71
Mixed spices	6	-	-	-	-	5
Salt	9	10	8	18	8.5	12
Sugar	35	32	36	45	1.5	5
KCl	0.5	10	9	9	8.5	6
Stevia powder (STV)	3	3	2	2	1.5	1
Dried basil leaves	-	-	-	7	-	-

G means green curry, P means Panang curry, M means Massaman curry, SB means spicy basil leaf curry, SS means southern sour curry, SY means southern spicy yellow curry, and STV means stevia powder.

## Data Availability

All data are contained within the article and Appendix A.

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
