# Peer review of "Antioxidant and Nitric Oxide Inhibitory Activity of the Six Most Popular Instant Thai Curries"

_foods, 2024, doi:10.3390/foods13020178_

Round 1

Reviewer 1 Report

Comments and Suggestions for Authors

I read carefully the paper “Antioxidant and nitric oxide inhibitory activity of the six most popular instant Thai curries” and I found it enough interesting especially I found a very good and detailed M&M paragraph.

Results should be improved, expecially the contributes from LC-MS results, they are very few pointed out along the results paragraph. I think they are important implementation on support the other antioxidant tests's results.

Along the text many time there are citations starting with [xx] number of the ref, maybe is better to write the name of the first author.

Other observations and/or questions are in the attached file, on highlighted word/sentences.

Author Response

Material and methods

  1. Along the text many time there are citations starting with [xx] number of the ref, maybe is better to write the name of the first author.
  • I already edited.
  1. Add the reason “why” modified 517 nm was used instead of 520 nm at DPPH radical scavenging activity.
  • I already rechecked and used 517 with changing the reference. In fact, flavonoids could be detected with 517 and 520 but 517 was fitted with our results due to less carotenoids interfering. (line 168)
  1. Adding the name of each compound detected or at least the class of compound could be useful at line 191.
  • I already edited as shown in line 206-208.

Results and discussion

  1. Delete “a high phenolic compound” at line 249.
  • I already deleted it.
  1. Could be better explain briefly why both range are so wide…(cultivars? analitic determinations) at line 262.
  • I already modified and stated that sample was collect from various local spice market in India as shown in line 277-279.
  1. Describe better your results and then add the reference at line 263-264.
  • I already edited as shown in line 280-282 due to 16 phenolic compounds detected by LC-MS were the result of Gurnani et al. [32].
  1. Your results on LC-MS should be more detailed at line 334-336.
  • I already edited as shown in line 379-382 due to 16 phenolic compounds detected by LC-MS were the result of Chaudhary et al. [46].
  1. Add the result on phenolic and flavonoid profiling by LC-MS.
  • Phenolic and flavonoid profiling by LC-MS was created in section 3.2 as shown in line 335-358.

  1. Results should be improved, expecially the contributes from LC-MS results, they are very few pointed out along the results paragraph. I think they are important implementation on support the other antioxidant tests's results.
  • Phenolic and flavonoid profiling by LC-MS was created in section 3.2 as shown in line 335-359.

Reviewer 2 Report

Comments and Suggestions for Authors

Subject of the manuscript is very interesting but there are lots of technical errors throughout manuscript that should be corrected, for example:

Line 95: It is suggested to use “°C” instead of “ oC ”; please correct accordingly throughout the text (e.g. Lines 97, 130, 131, 133 etc.)

How many repetitions were conducted per analysis?

Results should be presented in tables instead of figures for better clarity.

The authors should include in the interpretation of the results a PCA analysis that would show the dispersion of the results of the antioxidant activity by different determination methods (DPPH, FRAP, ABTS…)

Author Response

Material and methods

  1. Line 95: It is suggested to use “°C” instead of “ oC ”; please correct accordingly throughout the text (e.g. Lines 97, 130, 131, 133 etc.).
  • I already edited all.
  1. How many repetitions were conducted per analysis?
  • I already stated in line 251-252.

Result and discussion

  1. Results should be presented in tables instead of figures for better clarity.
  • After trying to change to tables, there was no better clarity, and it was difficult to compare. Therefore, I would like to present in figures.
  1. The authors should include in the interpretation of the results a PCA analysis that would show the dispersion of the results of the antioxidant activity by different determination methods (DPPH, FRAP, ABTS…)
  • It was a great comment, however there was no supported program available in Faculty and time limited to run a PCA in this work. But I will keep your suggestion in my team for further work and analysis.

Reviewer 3 Report

Comments and Suggestions for Authors

The problem undertaken at work entitled „Antioxidant and nitric oxide inhibitory activity of the six most popular instant Thai curries” is interesting. However, the manuscript requires significant improvement in terms of content, especially the introductory and conclusion part. Introduction should be rewritten focusing on background and novelty of presented work. 

The used modifications should be justified in the introduction.

Detailed comments

Regarding Aim of work:

L.66-72. There is no justification for study topic of the work, and a clearly formulated purpose of the manuscript. Moreover, it is unclear, what is the purpose of this work and what is information from the scientific literature.

Regarding Materials and Methods:

It is unclear how the authors determined the content of KCl and stevia powder. How many g of sugar/salt replaced 1 g of stevia/KCl? This information schould be added.

L.112- 114:  “Each dried curry powder was blended with seasoning using a mixer to obtain instant curry powder with particle size lower than 90 mesh.” - What type of mixer was used? How was the particle size measured? Also, SI units should be used in the work.

There is lack of information, about used statistical software. How many repetitions were performed?

Regarding Results and Discussion:

L.260: “[30] found that crude…” replace with: Gurnani et al. [30] found that crude …”. The comment refers to the total work.

Figure 9 is unreadable and should be corrected.

Regarding Conclusion:

In the summary part, major improve should be made. The summary does not present conclusions from this work. It also contains information typical for the work introduction. Conclusions, should include only conclusions from this work results.

Author Response

Introduction

  1. 66-72. There is no justification for study topic of the work, and a clearly formulated purpose of the manuscript. Moreover, it is unclear, what is the purpose of this work and what is information from the scientific literature.
  • I already edit as shown in line 67-79.

Materials and Methods

  1. It is unclear how the authors determined the content of KCl and stevia powder. How many g of sugar/salt replaced 1 g of stevia/KCl? This information schould be added.
  • I already edited as shown in line 127-131 however, some deep details were closed due to entrepreneur secret and petty patent.
  1. 112- 114: “Each dried curry powder was blended with seasoning using a mixer to obtain instant curry powder with particle size lower than 90 mesh.” - What type of mixer was used? How was the particle size measured? Also, SI units should be used in the work.
  • I already edited and changed it to SI units. (line 110, 117, 127)
  1. There is lack of information, about used statistical software. How many repetitions were performed?
  • I already edited as shown in line 253-254.

Results and Discussion

  1. 260: “[30] found that crude…” replace with: Gurnani et al. [30] found that crude …”. The comment refers to the total work.
  • I already edited all.
  1. Figure 9 is unreadable and should be corrected.
  • I already edited.

Conclusion

  1. In the summary part, major improve should be made. The summary does not present conclusions from this work. It also contains information typical for the work introduction. Conclusions, should include only conclusions from this work results.
  • I already edited in conclusion part line 493-517.

Round 2

Reviewer 3 Report

Comments and Suggestions for Authors

The manuscript is suitable for publication in Foods in its current form.